# Nonuniformity-Immune Read-In Integrated Circuit for Infrared Sensor Testing Systems

**Minji Cho [1], Heechul Lee [1] and Doohyung Woo [2],***

[1]   School of Electrical Engineering, Korea Advanced Institute of Science and Technology, Daejeon 34141, Korea; chominji2518@gmail.com (M.C.); hclee@kaist.ac.kr (H.L.)
[2]   School of Information, Communications and Electronics Engineering, The Catholic University of Korea, Gyeonggi-do 14662, Korea
*   Correspondence: cowpox@catholic.ac.kr; Tel.: +82-2-2164-4522

**Abstract:** In this study, a novel IR projector driver that can minimize nonuniformity in electric circuits, using a dual-current-programming structure, is proposed to generate high-quality infrared (IR) scenes for accurate sensor evaluation. Unlike the conventional current-mode structure, the proposed system reduces pixel-to-pixel nonuniformity by assigning two roles (data sampling and current driving) to a single transistor. A prototype of the proposed circuit was designed and fabricated using the SK-Hynix 0.18 μm CMOS process, and its performance was analyzed using post-layout simulation data. It was verified that nonuniformity, which is defined as the standard deviation divided by the mean radiance, could be reduced from 21% to less than 0.1%.

**Keywords:** sensor testing systems; infrared scene projector; current-programmed pixel; nonuniformity correction; read-in integrated circuit

---

## 1. Introduction

An infrared scene projector (IRSP) is a widely used piece of equipment that projects infrared (IR) images to investigate the performance of IR sensors [1–7]. These projectors are composed of IR emitters and driver integrated circuits (ICs) that are called read-in integrated circuits (RIICs) and are responsible for driving current to the IR emitters. High-quality IR scenes are required for accurately evaluating the performance of IR sensors; therefore, the IRSPs needs to be able to correct the nonuniformity between individual pixels of an IR emitter and an RIIC.

Currently, the look-up table (LUT) method is used to reduce the nonuniformity at each pixel [8–14]; however, as the radiance range of the overall pixels must be standardized to alower value to achieve uniform radiation, the performance of this method is hindered for a wide radiance range of IRSPs. Therefore, to obtain high-quality IR images for wide radiance ranges, a nonuniformity reduction in emitters and ICs is required in addition to external correction.

To improve circuit uniformities, several RIICs that adopt current-programming methods have been suggested [15,16]. However, to ensure the simultaneous IR emission of pixels [9,10], two transistors are separately needed for data sampling and current driving, hindering precise RIIC nonuniformity reduction.

In this paper, a novel RIIC design that enables precise nonuniformity compensation by adopting a dual-current-programming structure is proposed. In particular, with a mode-switching mechanism, a transistor can perform both data sampling and current driving, improving the uniformity of all pixels. Section 2 describes the implementation and analysis of the proposed nonuniformity-immune RIIC and its behavior is analyzed based on post-layout simulations presented in Section 3. Finally, conclusions are drawn in Section 4.

## 2. Proposed Nonuniformity-Immune RIIC

### 2.1. Nonuniformity Influences of the Conventional RIIC

The relationship between the current of the emitter, driven by the RIIC ($I_{emitter}$), and its final radiance ($\Gamma$) can be calculated using Equations (1) and (2) [17]:

$$T_{emitter} = \left(I_{emitter}^{2} \cdot R\right) \cdot G^{-1} + T_{sub}, \tag{1}$$

$$\eta(T) = \int_{\lambda_1}^{\lambda_2} 2hc^2 / \left[\lambda^5 \cdot \left(e^{\frac{hc}{\lambda k T_{emitter}}} - 1\right)\right] d\lambda \tag{2}$$

$$\Gamma = \eta\left(T_{app}\right) = \varepsilon \cdot ff \cdot \eta(T_{emitter}) \tag{3}$$

where $T_{emitter}$ and $T_{sub}$ are the temperatures of the emitter and substrate, respectively; $T_{app}$ is the apparent temperature considering the emissivity, $\varepsilon$, and fill factor, $ff$, of the IR emitter; $G$ and $R$ are the thermal conductance and resistance of the emitter, respectively. $\eta(T)$ represents Planck's equation; $\lambda_1$ and $\lambda_2$ are the wavelengths of interest; $h$, $k$ and $c$ are the Planck and Boltazman constants and the speed of light.

As indicated in Equation (1), $T_{emitter}$, used to determine pixel radiance in Equation (3), is proportional to the square of $I_{emitter}$. As a result, the control of the current by RIIC is critical to reduce the radiance gap between pixels. In this study, $R$, $G$, $\varepsilon$, and $ff$ were assumed equal to the ideal design parameters.

Figure 1a shows a block diagram of the RIIC. Every unit pixel of the IR display panel contains an IR emitter, represented as a resistor in Figure 1b. Conventional RIIC pixels contain two capacitors and a unity-gain buffer ($B_1$), responsible for a synchronized IR emission (snapshot operation), resulting in high-speed scene generation without defects [17–21]. However, as the RIIC layout area is limited by the pitch of the emitter, the small-sized $B_1$, shown in Figure 1b, is vulnerable to gain and offset errors that induce the nonuniformity of each pixel.

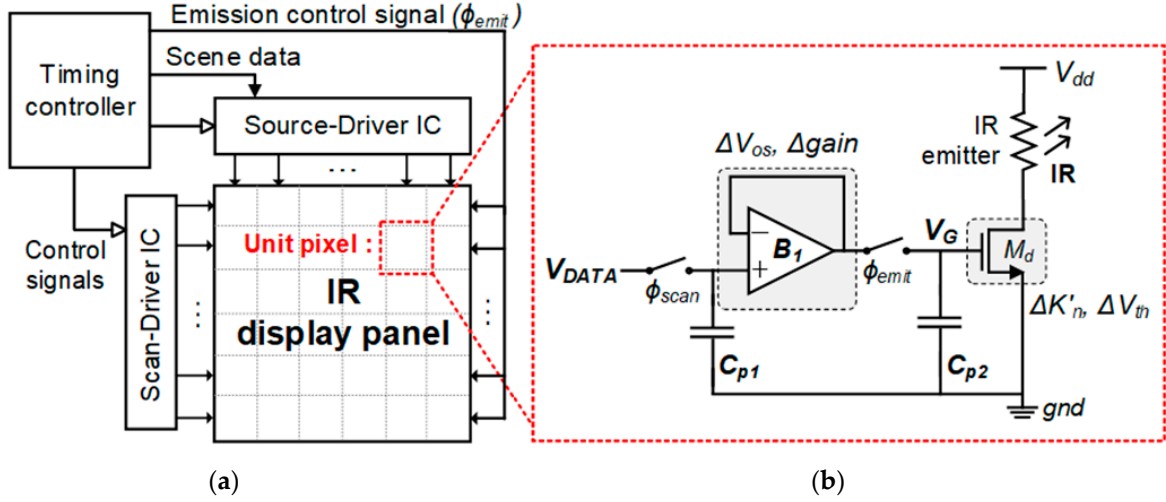

**Figure 1.** (**a**) Overall block diagram; (**b**) conventional read-in integrated circuits (RIIC) unit pixel.

For example, in Figure 2a, the dotted line $A_1$ represents a nonideal transfer curve affected by a gain and offset voltage ($V_{os}$) of the buffer. According to this curve, the voltage $V_{G1}$, applied to the gate node ($V_G$), is lower than that generated in an ideal situation, $V_{G0}$, at the same data voltage $V_{DATA0}$. This difference also affects the apparent temperature, as indicated by $T_0$ and $T_1$ in Figure 2b, resulting in pixel-to-pixel nonuniformity.

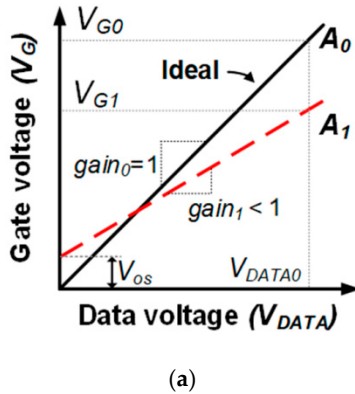
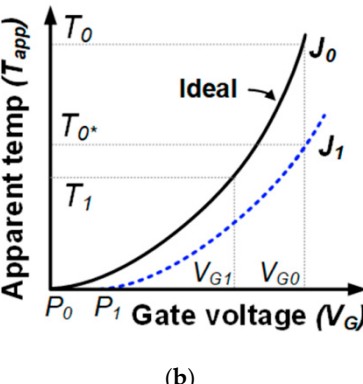

(a)                    (b)

**Figure 2.** Transfer curves: (**a**) data voltage against gate voltage; (**b**) gate voltage against apparent temperature ($A_1$, $J_1$: nonideal curves).

The relationship between $I_{emitter}$ and $V_G$ can be represented using the following equation:

$$I_{emitter} = \frac{1}{2} \cdot K_n \cdot \left(\frac{W}{L}\right)(V_G - V_{th})^2 = K'_n(V_G - V_{th})^2, \tag{4}$$

As shown in Equation (4), transistor mismatch parameters, the constant $K_n$ ($K_n = \mu_n \cdot C_{ox}$) and the threshold voltage $V_{th}$, result in the nonuniformity of $I_{emitter}$. As a result, even if the gate voltage is an ideal value $V_{G0}$, the apparent temperature can be a nonideal value $T_{0*}$, as indicated in $J_1$ in Figure 2b. The $x$-axis intercept $P_1$ and the slope of $J_1$ are affected by the difference in $V_{th}$ and $K'_n$, respectively. Therefore, a novel circuit design that avoids buffer errors and current-driving transistor mismatch is desired.

### 2.2. Proposed Nonuniformity-Immune RIIC

The proposed circuit comprises two current-driving transistors ($M_{d1}$ and $M_{d2}$), two capacitors ($C_{m1}$ and $C_{m2}$), and six switches (S1–S6), as shown in Figure 3a. The IR emitter is represented as a resistor in Figure 3a. Furthermore, $I_{data}$ represents the current-type scene data, which can be designed using a current-output digital-to-analog converter or a current-output source-driver [15,16]. Meanwhile, the pixel circuit comprises a dual-current-programming structure. The first is composed of $M_{d1}$, $C_{m1}$, S1, S3, and S5, while the other is formed by $M_{d2}$, $C_{m2}$, S2, S4, and S6. In the timing diagram, Figure 3b, two operating phases, whose period is one frame time, continuously repeat.

In phase I, the $I_{data}$ passes through S1, S3, and $M_{d1}$. As a result, the gate-source voltage of $M_{d1}$ ($V_{gs1}$) is sampled in $C_{m1}$ and can be calculated using the following equation:

$$V_{gs} = \sqrt{I_{data} \cdot \left(K'_{n1}\right)^{-1}} + V_{th1}, \tag{5}$$

Unlike voltage-programmed pixels, the $V_{gs}$ of the current-programmed pixel contains the mismatch parameters of $M_{d1}$. In phase II, the S5 switch turns on, enabling $M_{d1}$ to drive the current ($I_{emitter}$) to the emitter. In this stage, $I_{emitter}$ can be calculated using the following equation:

$$I_{emitter} = K'_{n1}\left(V_{gs} - V_{th1}\right)^2 = I_{data}, \tag{6}$$

It should be noted that the mismatch parameters of $M_{d1}$ are canceled when $V_{gs}$ from Equation (6) is substituted into Equation (5). As a result, when neglecting the channel-length modulation effect, the circuit can drive the emitter current regardless of the current-driving transistor mismatch. It should be noted that the Equations (4)–(6) are based on the operation in the saturation region; $M_{d1}$ and $M_{d2}$ need to operate in the saturation region. Thus, when designing the unit RIIC cell, the resistance of the

IR emitter and the amount of current driven to the emitter needs to be considered for guaranteeing the drain-to-source voltages ($V_{ds}$) of $M_{d1}$ and $M_{d2}$.

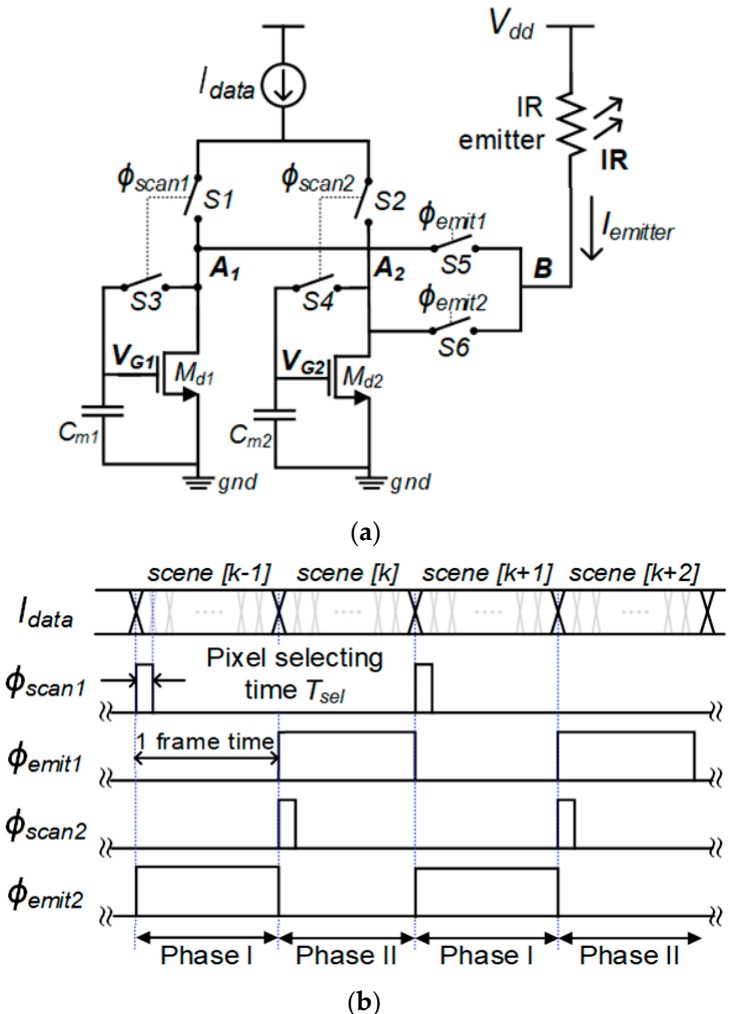

(a)

(b)

**Figure 3.** (**a**) Proposed RIIC unit pixel diagram; (**b**) corresponding timing diagram.

$M_{d2}$ samples the data voltage in phase II and drives the current to the subsequent phase I. As a result, $M_{d1}$ and $M_{d2}$ can operate in sampling or driving modes, enabling a snapshot operation without a unity-gain buffer $B_1$ and precise mismatch compensation. For example, in phase I, $M_{d1}$ is in the sampling mode and $M_{d2}$ is in the driving mode.

Figure 4 shows the circuit diagram between the $A_1$ and $B$ nodes in Figure 3a. In the sampling mode, when $S3$ is turned on, the drain-source voltage of $M_{d1}$ ($V_{ds1}$) is identical to the gate-source voltage of $M_{d1}$ ($V_{gs1}$). However, in the driving mode, the $V_{ds1}$ changes according to $V_{dd} - I_{emitter} \cdot R$, generating an unexpected charge ($\Delta Q$) at the gate node ($V_{G1}$). This generated error voltage ($V_{error,gs1}$) can be estimated using Equation (7), derived from the charge conversion law [22]:

$$V_{error,gs1} \cong C_{ov1}/C_{m1} \cdot \left(V_{ds1[driving\ mode]} - V_{ds1[sampling\ mode]}\right), \tag{7}$$

The error voltage also results in a current error, which can prevent accurate IR scene generation. Furthermore, the cascode structure, which is commonly used to neglect the effect of $\Delta V_{ds}$, cannot be used in the pixel as it prevents the dual functionality of $M_{d1}$ and $M_{d2}$. To avoid the existence of $\Delta Q$, a unity-gain buffer ($B_{p1}$) is inserted, as shown in Figure 4b. In particular, $B_{p1}$ is placed in a feedback loop during the sampling mode, preventing the pixel-to-pixel nonuniformity due to buffer errors.

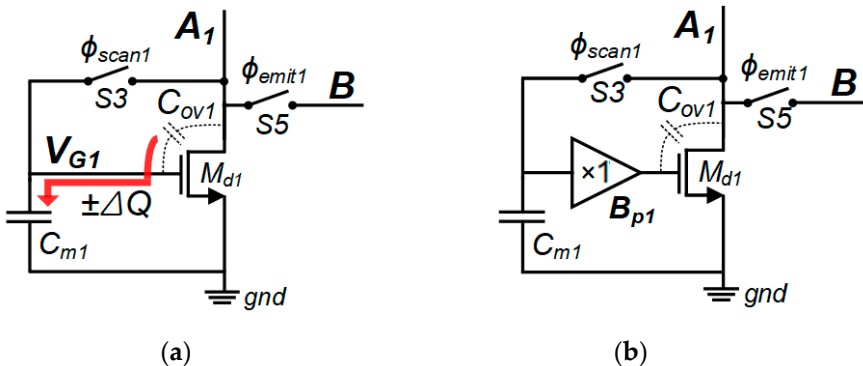

**Figure 4.** Buffer addition to improve data sampling accuracy: (**a**) pixel design without a buffer; (**b**) pixel design with a buffer.

Figure 5 shows the mask layout of the proposed RIIC pixel with two buffers ($B_{p1}$, $B_{p2}$) in a 56 μm pitch. The W/L parameters of $M_{d1}$ and $M_{d2}$ are both 10/10. Two pads are located in each pixel to establish a connection with the IR emitter device. Considering the symmetry of the layout, $C_1$ is divided into two subcomponents ($C_{1a}$ and $C_{2b}$) whose sizes are 150 fF and 600 fF, as shown in Figure 5. $C_2$ is analogously divided into $C_{2a}$ and $C_{2b}$.

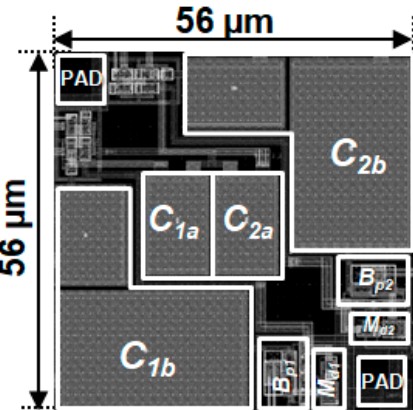

**Figure 5.** Mask layout of the proposed unit pixel in a 56 μm pitch.

## 3. Simulation Results and Analysis

### 3.1. Data Sampling Accuracy

To verify the feasibility of the proposed circuit, a prototype RIIC was designed via an SK-Hynix 0.18 μm CMOS process and apost-layout simulation on Cadence software (Cadence Design Systems, Inc., San Jose, CA, USA) was performed. The data current ($I_{data}$) varied from 0.1 to 200 μA, and the measured gate-source voltage of $M_{d1}$ varied as plotted in Figure 6. To verify the operating speed, the $I_{data}$ varied by one frame time 1/100 Hz = 10 ms, and the pixel-selecting time ($T_{sel}$) shown in Figure 3b was set to 1/(100 × 64 × 64) = 2.4 μs.

From the simulation results, the gate voltage error ($V_{error,gs1}$ = 2.5 mV) disappeared by using the $B_{p1}$, as indicated by the solid line in Figure 6b. Therefore, the unity-gain buffer helps maintain the sampled voltage regardless of the operational phase of the circuit.

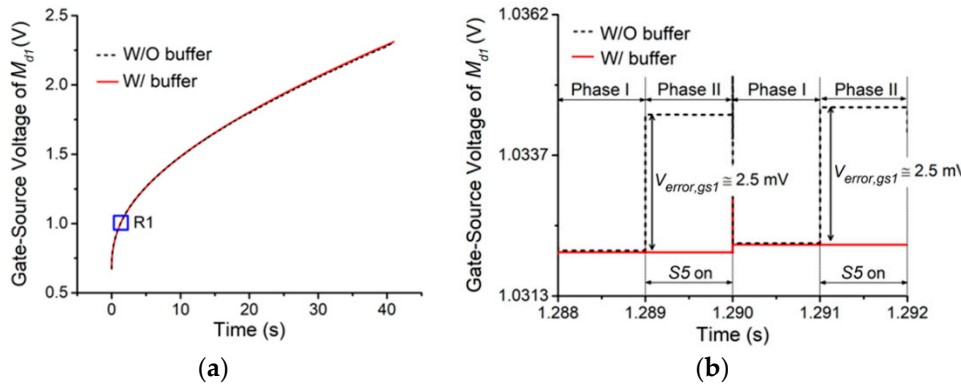

(a)　　　　　　　　　　　　　　　(b)

**Figure 6.** Post-layout simulation results: (**a**) gate-to-source voltage of $M_{d1}$ while varying the current data from 0.1 to 200 μA; (**b**) magnified view of region R1.

### 3.2. Evaluation of Nonuniformity

To evaluate the improvement in pixel-to-pixel uniformity, deviations in some quantities were purposefully inserted during the post-layout simulations. The offset voltage and gain error of $B_1$, which were estimated based on the mismatch parameters of the CMOS process, were set to 16 mV and 2%, respectively. The standard deviation ($\sigma$) of the threshold voltage was set to 3 mV following the data sheet of the CMOS process. Furthermore, the widths and lengths of $M_d$, $M_{d1}$ and $M_{d2}$ were identical. Moreover, the maximum current of the conventional RIIC was defined, according to the proposed RIIC, as 200 μA. This value is sufficient for the circuit to achieve the target temperature range listed in Table 1. Figure 7a,b shows the acquired emitter current data driven by the conventional RIIC and proposed RIIC, respectively.

**Table 1.** Target infrared scene projector (IRSP) properties.

| Properties | Value |
| --- | --- |
| Operating speed | 100 Hz |
| Array size | 64 × 64 |
| Pixel size | 56 μm × 56 μm |
| Digital input depth | 12 bits |
| Apparent temperature range | 275–700 K |
| Nonuniformity | <3% |

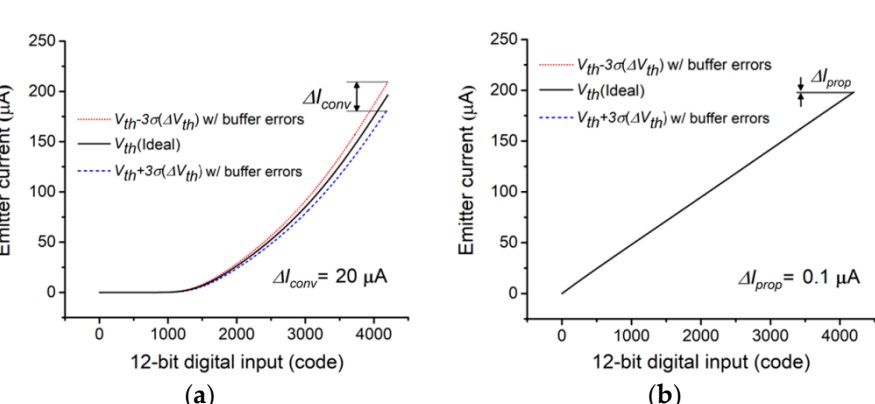

(a)　　　　　　　　　　　　　　　(b)

**Figure 7.** Emitter current of the IR emitter: (**a**) conventional RIIC; (**b**) proposed RIIC.

Using the current data acquired from the simulation, the apparent temperature was calculated using Equations (1) and (2): the desired properties of the IR emitter used in this calculation are listed in Table 2.

**Table 2.** Target IR emitter properties.

| Properties | Value |
| --- | --- |
| Resistance | 15 kΩ |
| $\varepsilon \cdot ff$ | 0.47 |
| $G$ | 1.0 μW/K |
| $\lambda_1, \lambda_2$ | 3 μm, 5 μm |

Figure 8 shows a graph of the relationship between digital input and apparent temperature considering the conventional and proposed RIIC, respectively, and Figure 9 shows a graph of the relationship between digital input and in-band power radiance. Comparing Figure 8a,b, the temperature difference between pixels decreases when using the proposed circuit structure. In particular, the maximum temperature difference decreased from 112 to 0.5 K.

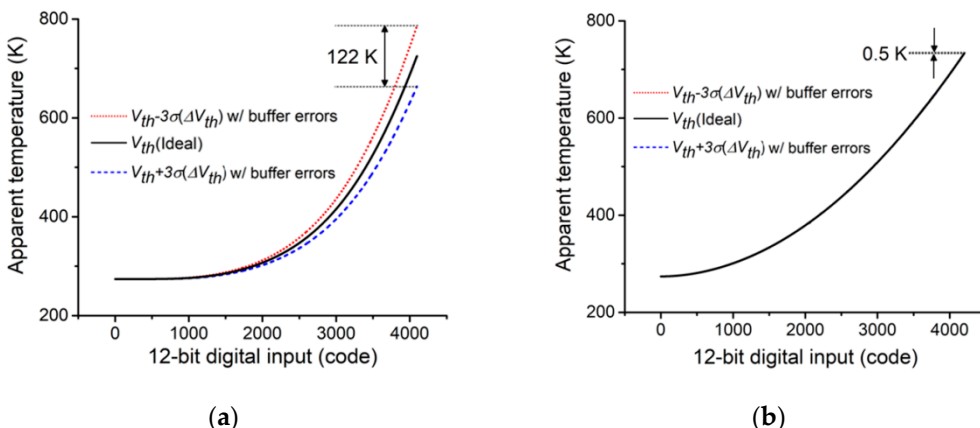

(**a**)          (**b**)

**Figure 8.** Apparent temperature analysis of IR emitter: (**a**) conventional RIIC; (**b**) proposed RIIC.

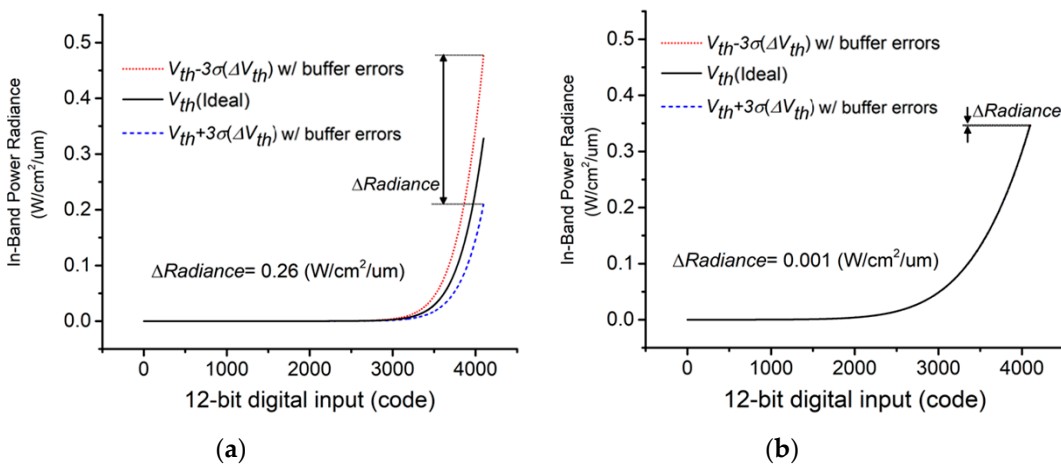

(**a**)          (**b**)

**Figure 9.** Radiance analysis of IR emitter: (**a**) conventional RIIC; (**b**) proposed RIIC.

The nonuniformity, which can be calculated with Equation (8) [9], was plotted as a function of the apparent temperature, as shown in Figure 10.

$$Nonuniformity(\%) = (\sigma_{radiance}/avg_{radiance}) \times 100, \tag{8}$$

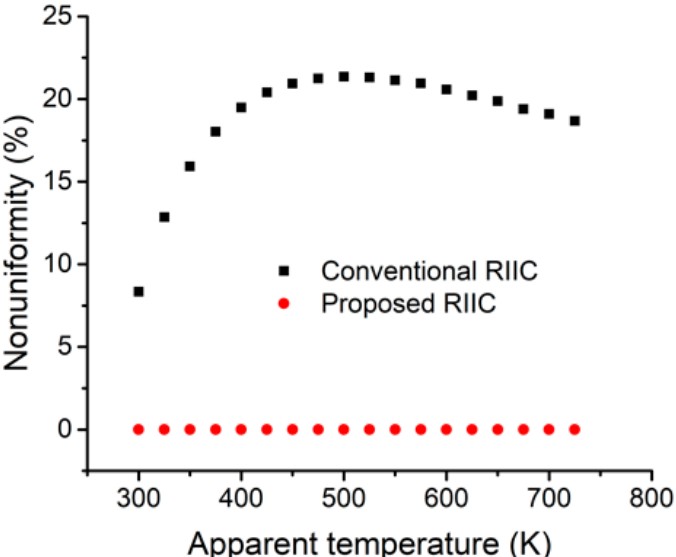

**Figure 10.** Nonuniformity comparison of IR emitter driven by the conventional and proposed RIICs.

The highest *nonuniformity* value for a conventional RIIC was 21%, however, when the proposed structure was used, this quantity became lower than 0.1%. This value, therefore, is sufficiently lower than the target specification listed in Table 1 (<3%), provingthe feasibility of the novel circuit through simulation and data analysis.

If the proposed circuit, however, was applied to a large-sized array, e.g., over 512 × 256 and operating at 200 Hz, accurate data sampling within a pixel-selecting time of nearly 0.04 µs would be needed [20]. As a result, the slow driving speed of the current-programming mechanism from the large line capacitance can prevent the performance of the proposed system [21–25]. In this case, the use of a column digital-to-analog converter (DAC) [26–30], which ensures a pixel-selecting time nearly 510 times longer than that of the single DAC structure, is a suitable solution.

## 4. Conclusions

We evaluated the operation and performance of a novel nonuniformity-immune RIIC composed of a dual-current-programming structure, which uses dual-functional transistors responsible for data sampling and current driving, to improve the accuracy of nonuniformity corrections. The results obtained from a post-layout simulation and data analyses indicated that the variance of the maximum apparent temperature and radiance nonuniformity were reduced from 122 to 0.5 K and from 21% to less than 0.1%, respectively. Therefore, the proposed RIIC design could be applied to IR sensor testing and evaluation applications with uniform IR scenes.

**Author Contributions:** D.W. supervised the research. M.C. and H.L. proposed the idea and designed the circuit. M.C. performed the simulation and analysis. M.C. and H.L. wrote the initial manuscript. D.W. revised and finalized the paper. All authors have read and agreed to the published version of the manuscript.

**Funding:** This research was funded by the Korea government (MSIT), grant number 2020R1F1A1052571.

**Acknowledgments:** This work was supported by the National Research Foundation of Korea (NRF) grant funded by the Korea government (MSIT) (No. 2020R1F1A1052571). This work was supported by the Catholic University of Korea, Research Fund, 2020. The authors are thankful to the IC Design Education Center in Korea for the support of design software.

**Conflicts of Interest:** The authors declare no conflict of interest.

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
