# Peer review of "Nonuniformity-Immune Read-In Integrated Circuit for Infrared Sensor Testing Systems"

_electronics, doi:10.3390/electronics9101603_

Round 1

Reviewer 1 Report

In this manuscript, the authors proposed a new design of read-in integrated circuit using two sets of current driving transistors. This design is expected to provide more accurate infrared irradiance than the traditional design. I find this manuscript well written, and the design has a potential market. There a few things that I want the authors to address or clarify before the manuscript can be accepted for publication.

(1) In Eq(2), does it refer to spectral resolve irradiance, or just integrated irradiance? The Planck's equation gives different spectral profile as a function of temperature.

(2) In the traditional RIIC, it is driven by voltage, i.e. V_data in Fig. 1(b), but in the proposed design, it is driven by current, i.e. I_data in Fig.  3(a). Why there is such a difference?

(3) How the 6 switches in Fig. 3(a) are controlled, what is the logic for switching them on and off? It only mentions that S1 and S3 are open in Phase I, and S5 is on in Phase II

(4) In section 3, it seems all the results are based on calculations. Therefore, the section title should be simulation or calculation results instead experimental results.

(5) The authors emphasize the improvement of nonuniformity from 20% to 0.1%. Can the author also present the irradiance as a function of digital input as in Fig.7 and 8?

Reviewer 2 Report

I appreciate the reading as the work has been clearly described.

Anyway, results are only related to simulation. This makes the work weak, even if theoretically seems to be interesting and valid.

Any comparison with the state-of-the-art, through a comparison table) is missing. Please specify if the results are related to measurements or simulation.

Please specify more the details regarding the read-out system (i.e. which DACs have been used). The paper is 9 pages only and it could report much more details. Please frankly describe the weakness of your approach compared to traditional approaches.

Please describe more about the post-layout simulations that have been performed (which tool? cadence? or someone more specific for your purpose?)

Reviewer 3 Report

  1. the presentation of the circuit schematic
    1. figure-1/3/4, perhaps a ground (analog) would help to clarify if VG, and Vgs have the similar meaning, and there is no body effect  
    2. it also helps explain equation 3, which is in saturation
    3. it is also beneficial to the reader to explain how to ensure the transistor Md operates in saturation region
  2. 0.18 um process or "0.18 um CMOS process"? or this is not a CMOS circuit?
  3. how to implement buffer Bp1? is it a source follower? it would be fine with any circuit implementation due to low frequency.
  4. it is short fall that no comparison to existing works.
